# Molecular Evidence of *Rickettsia conorii* subsp. *raoultii* and *Rickettsia felis* in *Haemaphysalis intermedia* Ticks in Sirumalai, Eastern Ghats, Tamil Nadu, South India

**DOI:** 10.3390/microorganisms11071713

**Published:** 2023-06-30

**Authors:** Krishnamoorthy Nallan, Veerapathiran Ayyavu, Elango Ayyanar, Balaji Thirupathi, Bhavna Gupta, Panneer Devaraju, Ashwani Kumar, Paramasivan Rajaiah

**Affiliations:** 1ICMR-Vector Control Research Centre, Field Unit, 4. Sarojini Street, Madurai 625002, India; krishnamoorthy.n@icmr.gov.in (K.N.); balaji.t@icmr.gov.in (B.T.); bhavna.g@icmr.gov.in (B.G.); 2ICMR-Vector Control Research Centre, Puducherry 605006, India; elango.ayyanar@icmr.gov.in (E.A.); panneer.d@icmr.gov.in (P.D.); ashwani.kumar@icmr.gov.in (A.K.)

**Keywords:** *Rickettsia raoultii*, *Rickettsia felis*, *Haemaphysalis intermedia*, DNA Barcode, spotted fever group (SFG), *Rickettsia*, MLST, tick borne pathogens (TBP)

## Abstract

*Rickettsia* is an important pathogenic entity among tick-borne diseases (TBD), which are considered serious emerging public health problems globally. In India, though the widespread distribution of ticks and TBD has been documented, its real burden remains underreported. In a preliminary attempt, rickettsial surveillance was carried out in ticks collected from Sirumalai, Eastern Ghats in Tamil Nadu, India by using pathogen genome-based phylogenetic inferences generated through multi-locus sequence typing (MLST), targeting the genes 16s rRNA, OmpA, OmpB, and gltA by nested PCR. The laboratory evidence confirms the circulation of *Rickettsia* in *Haemaphysalis intermedia* species collected from this area. Analysis of the four gene sequences detected demonstrates their closest identity to the spotted fever group (SFG) available in the GenBank database. Further, multiple sequence alignment with other sequences derived from the GenBank database showed close relatedness to *Rickettsia conorii* subsp. *raoultii* (16s rDNA-99.32%, OmpA-93.38%, OmpB-97.39%, and gltA-98.57%) and *Rickettsia felis* (16s rDNA 99.54%, OmpA-100%, OmpB-100% and gltA-99.41%). With this genomic evidence, the circulation of rickettsial pathogens in the pools of *H. intermedia* ticks infesting livestock in the Sirumalai foothill area has been demonstrated and to complement the microscopic identification of the tick species, DNA barcodes were generated for *H. intermedia* using the mitochondrial cytochrome c oxidase subunit I gene (COI). Nevertheless, *R. raoultii* and *R. felis* were found to be the aetiological agents of tick-borne lymphadenopathy and flea-borne spotted fever in human cases, respectively, further study on the determination of their diversity, distribution, clinical relevance, and potential risk to the local community in these areas is highly warranted.

## 1. Introduction

Emerging and re-emerging tick-borne diseases (TBD) are reported to be on the rise globally [1,2] many of them are known to be of potential zoonotic origin [3]. Due to the recent global warming phenomenon, the expanding potential of the geographical distribution of ticks has also increased [4,5]. The incursion of exotic tick fauna into the naïve areas determines the introduction and spreading of emerging and remerging tick-borne pathogens (TBP) which necessitates the routine surveillance, estimation, and understanding of risk factors [6]. Even though the estimation of the tick-borne rickettsial disease burden is difficult due to misdiagnosis, its infection in Southeast Asia is estimated to be the second most reported next to malaria [7]. The rickettsial bacteria are mainly transmitted by the bite of infected ticks to rodents, dogs, and wild animals, as well as humans engaging in outdoor activities.

The genus *Rickettsia* is genetically subdivided into four groups based on molecular analysis, such as the spotted fever group (SFG), the typhus group, *Rickettsia bellii* group, *and Rickettsia canadensis* group [8]. SFG includes the causative agent of *R. rickettsii, R. conorii, R. africae, R. parkeri, R. honei, R. felis, R. conorii* subsp. *raoultii*, *R. conorii* subsp. *indica* and many others. Tick-borne rickettsioses are mainly caused by the SFG; its identification at the species level with immunological methods is difficult since ticks are a more competent and versatile vector [9]. The studies on spotted fever group (SFG) and typhus group (TG) rickettsioses are limited when compared with Scrub typhus (ST) [10]. However, worldwide, improved diagnostic methods have contributed to the reliable identification of tick-borne pathogens (TBP) in humans, domestic animals and vectors [11].

In India, the distributions of 106 tick species have been documented and only a few of them have been incriminated as potential vectors of known TBP [12]. Apart from its significant role in the transmission of viruses, *Rickettsia* remains one of the most important tick-transmitted etiology of causing rocky mountain spotted fever, rickettsialpox, other spotted fevers, epidemic typhus, and murine typhus in humans and animals in Southeast Asia (SEA) [13,14]. However, their true magnitude in terms of disease prevalence is yet to be determined [13]. Rickettsial isolates were recorded from ticks (*Amblyomma* spp., *Haemaphysalis* spp. and *Rhipicephalus* spp.) and fleas (*Ceratophyllus fasciatus* (rat flea) and *C. orientis felis*). In addition, molecular studies conducted Tamil Nadu, the north-eastern state of Nagaland and Punjab have also reported rickettsial infection in humans [15,16,17]. Our recent study has recorded the dominant occurrence of *Haemaphysalis intermedia* in certain parts of Tamil Nadu (unpublished data) and has already been known to transmit many public health important viruses, namely the Nairobi sheep disease virus in Sri Lanka, Bhanja and the Ganjam viruses in Odisha state, India [18,19].

Though outbreaks due to rickettsial etiology have been well documented in many Indian states [20], the available data on the circulation of *Rickettsia* through serological and molecular evidence and their clinical relevance in causing illness in humans is very scanty, but this evidence confirms its widespread distribution in the country [21,22,23]. The recent reports of encephalitis cases due the rickettsial infection spread by ectoparasites stress attention to molecular diagnosis SFG and TG because only >10% outbreaks of acute encephalitis syndrome (AES) etiology were confirmed and the reaming is not confirmed with molecular methods [10].

The data on the prevalence of rickettsial pathogens in ticks from the southern parts of Tamil Nadu, India is limited and no substantiated records are available from the Eastern Ghats. In addition, this area is known to have a favorable climate and rich biodiversity suitable for the breeding and spread of ticks and tick-borne diseases. Considering the emerging potential of tick-borne pathogens in India, the present study was conducted as a pilot laboratory-based screening of tick-borne rickettsial organisms in dominant tick species of *H. intermedia*, through PCR based multilocus sequence typing (MLST) [24]. Since this method is robust, eliminates phylogenetic inconsistencies due to lateral gene transfer and is widely used as a complementary tool for the identification of rickettsial species [25].

## 2. Materials and Methods

### 2.1. Study Area

The Sirumalai foothill ranges cover approximately 60,000 acres and are located in the last mountain range of the Eastern Ghats, Tamil Nadu, South India, at longitude: 77.99, latitude: 10.19, and elevation: 1600 meters (Figure 1). The villages situated in the Sirumalai foothills area geographically have a dense forest with a moderate climate (14 °C to 30 °C) throughout the year. The village population is mainly engaged in agricultural activities and cattle rearing. Livestock such as cows, sheep, goats, dogs and horses are domesticated in the villages. The prevailing climatic and ecological conditions are highly favorable for the breeding of ticks in this area.

### 2.2. Tick Collection and Identification

Tick specimens were collected from cows, goats, and dogs manually, using forceps, from October to December 2020. The individual specimens were immediately placed into a 2 mL vial containing 70% ethanol and transported to the laboratory for further processing. The samples were morphologically identified using standard keys [26,27] and pooled by sex and stage.

### 2.3. DNA Extraction

#### 2.3.1. *Rickettsia* Screening

Tick samples were removed from 70% ethanol (EtoH) washed twice in clean de-ionized water to remove the residue of EtOH and allowed to air dry before DNA extraction. Before the commencement of the homogenization, single female ticks, pools of males and other stages (5 ticks per pool) kept in 1.5 mL microtubes were incubated at 80 °C in a dry bath for 2 h to remove the moisture contents in the sample. All the microtube lids are kept open during the incubation to avoid the formation of moisture in the sample tubes. After complete drying, the whole body of samples in 1.5 mL microtubes was manually ground using the sterile stainless steel pellet pestle (Sigma-Aldrich, St. Louis, MO, USA, Subsidiaries: Supelco, In cat No. Z359963) without adding ATL buffer, until the tick was completely ground to powder. After manual grinding, 50 µL of the ATL buffer was added to each tube and homogenized with a motor tissue homogenizer (REMI-RQ-127A/D, Mumbai, India) at the optimum speed for 2 min using an individual pestle for each pool. Finally, the pestles were washed with 130 µL of ATL buffer (a total of 180 µL).

Since the *H. intermedia* females are bigger than the males and other stages, the adult female ticks were homogenized individually and after centrifugation for 1 min at 8000 rpm, the supernatant of each tube was collected and pooled into a single tube (five female ticks constitute a pool). On the other hand, males, larvae and nymphs were pooled as five specimens per pool for homogenization and after centrifugation, the supernatant was collected into fresh microtubes. All the supernatant was incubated at 56 °C in a dry bath for 1 h with 20 µL proteinase-K after a brief vortex. The samples were removed from a dry bath after one hour of protease digestion and column-based DNA extraction protocol was followed as per the manufacturer’s instructions (QIAamp DNA mini kit (Qiagen, Hilden, Germany)). The DNA was eluted with 50 µL of nuclease-free water (Qiagen, Cat. No. 129115) and stored at −80 °C.

#### 2.3.2. DNA Barcoding of *H. intermedia*

DNA was extracted from *H. intermedia* male and female ticks after morphological identification with standard keys [26,27]. Each tick specimen was cut into two parts vertically, half of the specimen was used for DNA extraction and the remaining half was preserved in the VCRC tick museum as a voucher specimen for future reference (Figure 2 e,f). The similar DNA extraction procedure for screening of *Rickettsia* was followed, except that the collection of the supernatant after centrifugation for 1 min at 8000 rpm was skipped; instead, the entire homogenate was used for protease digestion and the manufacturer’s protocol for column-based DNA extraction was followed (QIAamp DNA mini kit, Qiagen, Hilden, Germany). The DNA was eluted with 30 µL of nuclease-free water (Qiagen, Cat. No. 129115) and stored at −80 °C.

### 2.4. PCR Methods

#### 2.4.1. Multilocus Sequence Typing (MLST)

The total DNA extracted from each pool was screened for rickettsial species by targeting the OmpB gene using the nested PCR method. The positive samples for OmpB were further subjected to multi-locus sequence typing (MLST) by covering the genes encoding 16s rRNA, OmpA and gltA with gene-specific primers. Different types of PCR methods, such as single-stage, semi-nested and nested PCR, were carried out as per the standardized thermal cycler conditions for amplification of the four genes (Table 1) using the primer sequences for the detection of rickettsial species [28]. Both the first and second rounds of each PCR were carried out in a final volume of 25 µL. Two µL of template DNA was used for the first round of PCR and one µL of the first PCR product was used as a template for the second round of nested PCR. Six µL of PCR product was used for 1.5% agarose gel electrophoresis and after the gel-documentation, the remaining PCR amplified product was purified using the QIAquick PCR Purification Kit (Qiagen, Hilden, Germany). The DNA concentration was measured using the NanoDrop™ Lite Spectrophotometer (Thermo Fischer Scientific, Waltham, MA, USA) and the concentration of the PCR product with more than 50 ng/µL was subjected to DNA sequencing in both forward and reverse directions with the same primers used for PCR amplification. Primers that are used for the second round of PCR amplification were used for DNA sequencing in the cases of nested-PCR and semi-nested-PCR assays.

#### 2.4.2. DNA Barcoding

Mitochondrial cytochrome c oxidase subunit I gene (COI) fragment of 710 bp was PCR amplified with a touchdown PCR protocol using forward and reverse primers LCO1490: 5′-GGTCAACAAATCATAAAGATATTGG-3′, HC02198: 5′-TAAACTTCAGGGT GACCAAAAAATCA-3′ as described by Folmer et al., [29]. Two µL of DNA was used as a template and PCR was carried out with a total volume of 50 µL, using the Taq PCR core kit (Cat. No. 201223, Qiagen, Hilden, Germany).

### 2.5. DNA Sequencing and Analysis

#### 2.5.1. Rickettsial DNA Sequences

Eleven fragments (16s rRNA- 2, OmpA-2, OmpB-4 and GltA-3) from four PCR-positive pools were subjected to the Sanger DNA sequencing method. The forward and reverse DNA sequences were aligned and manually edited compared with the sequencing chromatogram. The partial DNA sequences thus obtained were analyzed in the BLASTn nucleotide database (https://www.ncbi.nlm.nih.gov) to find out their pair-wise nucleotide similarity with the sequences available in the nucleotide database (NCBI). A phylogenetic tree was constructed using the sequences with a higher percentage of identity with the query sequence, applying the maximum likelihood (ML) method using the Kimura 2 parameter (K2P) distances with 1000 bootstrap replicates on the MEGA 7 software (https://www.megasoftware.net) [30]. The outlier group of DNA sequences for *R. raoultii* and *R. felis* was also included in the FASTA file in order to separate the phylogenetic clade and one DNA sequence from a genetically more distant species was incorporated to serve as an outgroup in the phylogenetic tree.

#### 2.5.2. *H. intermedia* COI Sequences

DNA barcodes generated to complement the taxonomic identification of tick using the mitochondrial cytochrome c oxidase subunit I gene (COI) of *H. intermedia* were aligned in MEGA 7 software [30] and the final sequences were analyzed in the NCBI, BLASTn database for confirmation of the tick species.

## 3. Results

### 3.1. Detection of Rickettsial Pathogen

Tick-borne rickettsial pathogens remain an important clinical entity in causing serious illnesses in humans and animals, which are mostly unreported or underreported. In the present study, *H. intermedia* males, females, larvae and nymph life stages were screened for rickettsial organisms. During the survey, 330 *H. intermedia* were collected from cows (females 10, males 57), goats (larvae 30, nymphs 70, females 30, males 131) and dogs (males 2) (Figure 2) subjected to rickettsial screening. Table 2 shows the number of pools processed in each life stage, by sex, the number of genes sequenced, the number of positives and the *Rickettsia* species detected from the respective pools of *H. intermedia* using MLST.

The outer membrane protein B (OmpB) was PCR amplified from a pool of females (*R. felis*) and two pools of males (*R. felis* and *R. raoultii*). The sequence similarities of OmpB with the previous DNA sequences were 97.39% and 100% for *R. raoultii* and *R. felis*, respectively (Figure 3a). In addition, one pool positive for OmpB in the nymph stage (437 bp) detected showed 94% query coverage and 99.76% DNA sequence identity with *R. massiliae* strain 80 (GenBank No. KJ663753) isolated from Ixodid ticks *R. sanguineus* infesting humans in Italy. Since the other three loci did not yield PCR amplification, the OmpB nymph stage positive sequence was not submitted in the GenBank database and this pool positive was considered as *Rickettsia* spp.

In the case of the 16s RNA gene, single-step PCR was carried out to amplify 426 bp of the *Rickettsia* genome. Among the four positive pools obtained, the rRNA gene sequence could be amplified only in two pools. The multiple sequence analysis in the BLASTn database showed that each belongs to *R. felis* and *R. conorii* supsp. *raoultii*. The nucleotide identity of the query sequence with *Rickettsia* sp. and *R. felis* was found to be 99.54%. Another pool had 99.32% and 99.08% sequence identity with *Rickettsia* sp. and *R. raoultii*, respectively (Figure 3b).

*Rickettsia* outer membrane protein-A was amplified in the female and male pools and the BLASTn analysis of *OmpA* DNA sequences showed 100% and 93.38% sequence identity with *R. raoultii and R. felis*, respectively (Figure 3c). The citrate synthase protein coding gene (gltA) was amplified from *R. felis* in female and male pools and from *R. raoultii* in a male pool. The BLASTn analysis showed 99.41% and 98.57% sequence identity, respectively. DNA sequences (OM675975, OM675976 and OK633271) of both pathogens generated in this study have shown a close similarity with rickettsial sequences submitted from China and USA (Figure 3d) (GenBank Acc. Nos. MG818715 and MT050445).

Overall, the nucleotide identity of the rickettsial species was confirmed based on the values of the BLASTn DNA sequence analysis, such as the e-value, the higher the percentage of query coverage and sequence identity. The cut-off value (%) as proposed by Fournier et al., was considered for rickettsial species identification. The details of the rickettsial DNA sequence generated from the present study and the output values of BLASTn analysis are given in Table 3.

### 3.2. DNA Barcoding of Haemaphysalis intermedia

To complement the taxonomical identification of *H. intermedia*, it was subjected to DNA barcoding and 710 bp of the mitochondrial COI gene was amplified using template DNA extracted from a half portion of a male and female tick specimen collected from the study area. DNA sequences generated (700 bp, 699 bp) were subjected to BLASTn analysis and the result showed close similarity with the species viz., *H. japonica* (MK863383), *H. bispinosa* (MW078971), *H. danieli* (NC_062065) and *H. hoodi* (ON191014) with 85% to 90% query coverage. Therefore, the COI sequences generated in this study were submitted to GenBank as *H. intermedia* for the first time (Figure 4) (Accession Nos. OQ946978 and OQ946979).

## 4. Discussion

Rickettsial diseases are considered important public health problems globally and ticks play a major role in transmitting them to humans and animals [1,2]. Despite their true burden among the global community is not known, the recent reports on the escalating case incidences of tick-borne infections stress an in-depth understanding of the occurrence of the pathogen and strengthening of surveillance on TBP around the globe [31,32]. In India, tick-transmitted rickettsial infections are very common, but their true health impact is either unreported or underreported [33,34]. In Tamil Nadu, the occurrence of different tick species has been well documented [35] and their impact in causing diseases needs to be determined. The present study reports the circulation of *R. felis* and *R. raoultii* among tick species that are prevalent in Sirumalai foothill villages in Tamil Nadu. Though the pathogens detected were not associated with any clinical diseases in India, they were found to be etiological agents of the diseases, namely scalp eschar and neck lymphadenopathy after a tick bite (SENLAT) [36]. Hence, it is noteworthy to determine its clinical relevance to local animals and humans in causing diseases and their impact on the local community.

In order to generate adequate molecular evidence of the detected rickettsial sequences, multi-locus sequencing typing was carried out to fulfill the recommended essential criteria for determining the species within the genus *Rickettsia* [24,37]. A set of four genes, namely 16 s rRNA, OmpA, OmpB and gltA sequences, confirmed its identity as *R. raoultii* and *R. felis* in *H. intermedia* ticks in the study area. At the same time, the problems of non-amplification of these four genes from all the positive pools obtained were noted. Though the loci *OmpB and gltA were* successfully amplified in all three positive pools, (OmpB: *R. felis* OM675973 -OM675974 and *R. raoultii* OK633270; gltA: *R. felis* OM675975, OM675976 and OK633271 *R. raoultii*), the genes OmpA (OK633269-*R. raoultii*, OM675977-*R. felis*) and 16 s rRNA (OP185249-*R. raoultii* and OP185250-*R. felis*) were amplified in two pools. In a pool of males, the occurrence of *R. felis* was confirmed based on three loci, viz., 16S *rRNA*, OmpB and *gltA* gene sequences. The reason behind the failure of amplification of the *OmpA* fragment may be due to the truncated nucleotides, ranging from 36 bp to 4.5 kb, as reported by Ellison et al., [38].

All four loci of *R. conorii* subsp. *raoultii* were amplified in one of the male positive pool. The rRNA, OmpA, OmpB and gltA gene sequences of R. conorii subsp. *raoultii* were found to be clustered with isolates from China, Russia, Tibet, Japan and the United States (Figure 3a–d). Similarly, in the phylogenetic analysis, *R. felis* partial gene sequences of rRNA, OmpA, OmpB and gltA were found clustered with rickettsial strains originating from Germany and Mexico (EU071486, MF303722), Mexico (AJ563398), Spain (MK301596) and Taiwan (MT847619), respectively, which denotes the widespread distribution of *Rickettsia* around the globe. The OmpB positive in a pool of nymph stage, confirm that the tick carries rickettsial bacteria in all developmental stage and the transstadial transmission of *Rickettsia* by ticks.

Though the available evidence documents the widespread distribution of *Rickettsia* in various places in India, with our limited knowledge, this is the first report on the molecular evidence with MLST-based identification of *R. felis* and *R. conorii* subsp. raoultii in the *H. intermedia* ticks from south India. To complement the taxonomical identification of the tick species, partial DNA sequences of the cytochrome oxidase subunit I (COI) gene were generated. Upon comparison with BLASTn database sequences, they showed close identity to tick species, viz., *H. japonica* (MK863383), *H. bispinosa* (MW078971), *H. danieli* (NC_062065) and *H. hoodi* (ON191014 with 85% to 90% query coverage. Eventually, the vouched specimens of *H. intermedia* were maintained at the institute for future examination. The identity of the species was also reexamined with taxonomists unanimously and the COI sequence generated in this study has been deposited for the first time as *H. intermedia* in the GenBank database (OQ946978 and OQ946979).

The report on *R. raoultii* isolation from *Dermacentor reticulatus*, *D. marginatus*, *D. silvarum* and *D. nuttalli* ticks [39] and *R. felis* from *Ctenocephalides felis* (Cat flea), L. *bostrychophila* (book louse), *R. sanguineus* (the brown dog tick), *Xenopsylla cheopis* (rat flea) and *Ixodes granulatus* (tick) [40] have already been documented in the literature. At the same time, the mere circulation of rickettsial pathogens in ticks in this area cannot be correlated with the transmission risk to humans without adequate laboratory evidence. However, their existence in these areas could be considered a risk factor in the future, since, in India, six episodes of rickettsial outbreaks have already been reported [20] including the involvement of the first human case of *R. felis* from the north-eastern state of Nagaland, India [16].

## 5. Conclusions

In conclusion, the present study has shown laboratory evidence for the circulation of *R. conorii* subsp. *raoultii* and *R. felis* in *H. intermedia* ticks in the Sirumalai area of Tamil Nadu, India. The MLST is very useful and complementary in identifying the rickettsial agent. Application of this tool will reveal the actual rickettsial species in circulation in the environment. Further studies are highly warranted to determine the extent of the circulation of these agents and their impact on the community. It is worth understanding their role in causing clinical diseases in animals and humans and the potential risks of acquiring the diseases for future prevention and control measures.

## Figures and Tables

**Figure 1 microorganisms-11-01713-f001:**
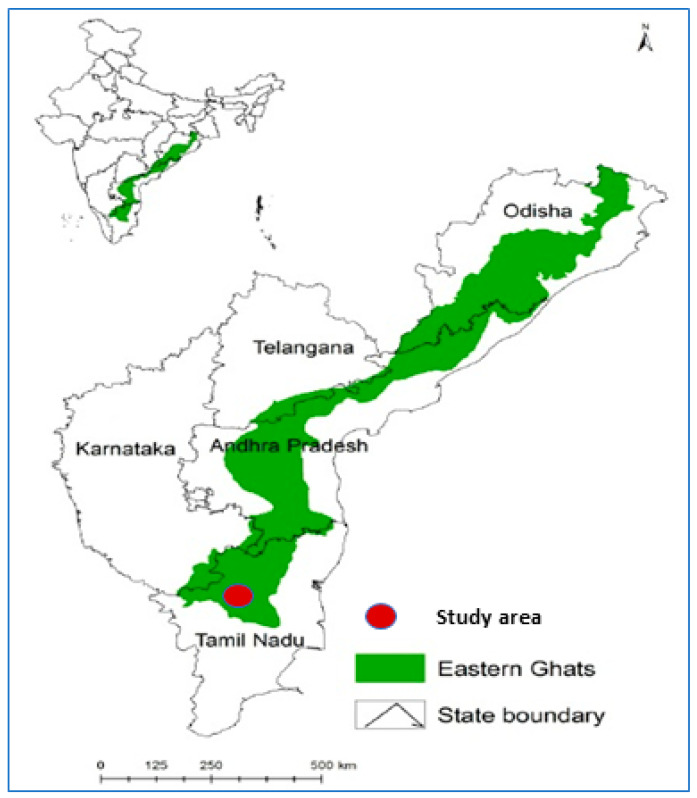
Map showing the study site located in the Eastern Ghats of Tamil Nadu, South India.

**Figure 2 microorganisms-11-01713-f002:**
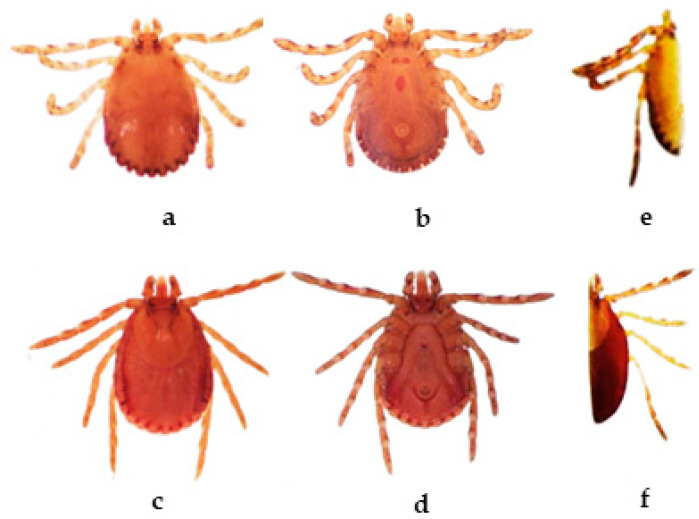
(**a**,**b**) Dorsal, ventral (male); (**c**,**d**) Dorsal, ventral (female); (**e**,**f**): Voucher specimens of *H. intermedia*.

**Figure 3 microorganisms-11-01713-f003:**
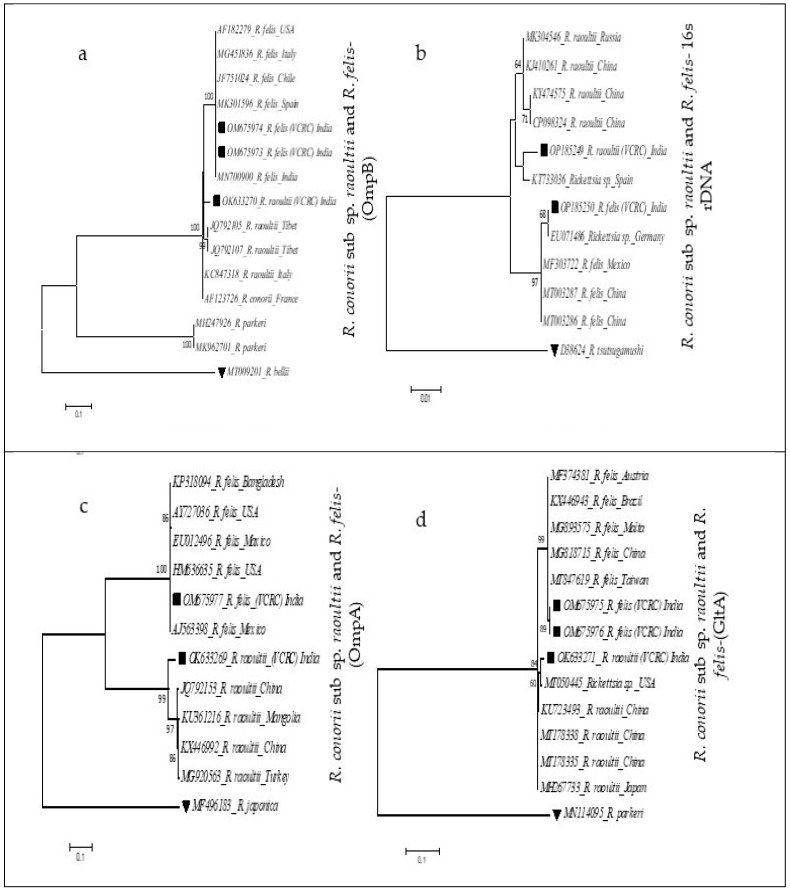
(**a**–**d**) Phylogenetic analysis of *Rickettsia* partial gene sequences of 16 s rDNA, OmpA, OmpB and GltA of *R. felis* and *R. conorii* subsp. *raoultii* using the maximum likelihood K2P model, nucleotide substitution with 1000 bootstrap replication in MEGA 7. Sequences of the closest neighbors and other rickettsial species from GenBank were incorporated to distinguish the phylogenetic glades. Legends: ▀ sequences generated in this study; ▼ Out-group.

**Figure 4 microorganisms-11-01713-f004:**
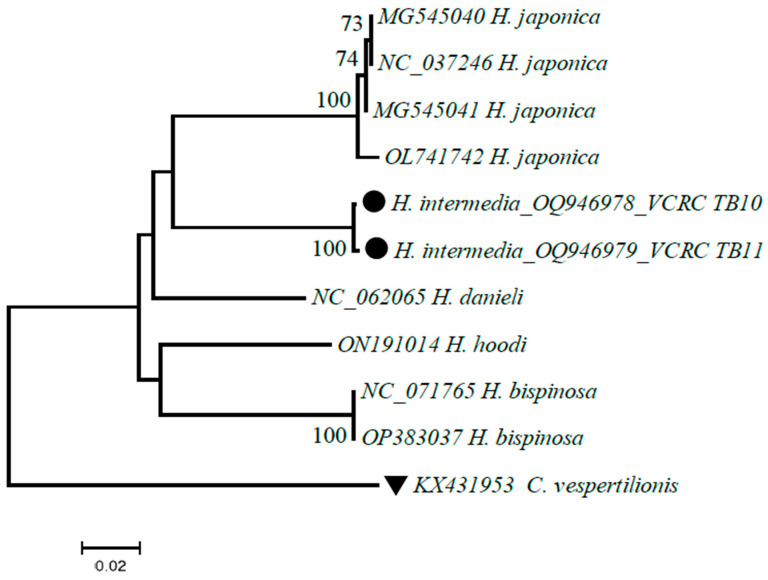
Phylogenetic analysis of partial COI gene sequences of *H. intermedia* using the neighbor-joining K2P model with 1000 bootstrap replication in MEGA 7. Sequences of the closest neighbors from the BLASTn analysis were incorporated in the phylogenetic analysis. Legends: ● sequences generated in this study; ▼ out-group.

**Table 1 microorganisms-11-01713-t001:** Details of the amplified genes, primer sequences, thermal cycler conditions and size of the DNA fragment.

Gene	Primer Name	Sequence 5′ to 3′	PCR Type	Thermal Cycler Conditions	Amplicon Size (bp)
*16s rDNA*	fD1 Rc16S.452n	AGAGTTTGATCCTGGCTCAG AACGTCATTATCTTCCTTGC	Single-stage	95 °C 4 min, 35 cycles (94 °C 1 min, 52 °C 50 s, 72 °C 1 min), 72 °C 7 min.	426
*ompA*	Rr190.70p Rr190.701n	ATGGCGAATATTTCTCCAAAA GTTCCGTTAATGGCAGCATCT	Semi- Nested	95 °C 4 min, 35 cycles (94 °C 1 min, 50 °C 50 s, 72 °C 1 min), 72 °C 7 min.	631
	Rr190.70p Rr190.602n	ATGGCGAATATTTCTCCAAAA AGTGCAGCATTCGCTCCCCCT		95 °C 4 min, 35 cycles (94 °C 1 min, 50 °C 40 s, 72 °C 50 s), 72 °C 7 min.	532
*ompB*	r*OmpB* OF r*OmpB* OR	GTAACCGGAAGTATCGTTTCGTAA GCTTTATAACCAGCTAAACCACC	Nested	95 °C 4 min, 35 cycles (94 °C 40 s, 52 °C 45 s, 72 °C 1 min), 72 °C 7 min.	511
	r*OmpB* SFG IF r*OmpB* SFG IR	GTTTAATACGTGCTGCTAACCAA GGTTTGGCCCATATACCATAAG		95 °C 4 min, 35 cycles (94 °C 40 s, 51 °C 45 s, 72 °C 1 min), 72 °C 7 min.	420
*gltA*	RpCS.877p RpCS.1258n	GGGGGCCTGCTCACGGCGG ATTGCAAAAAGTACAGTGAACA	Nested	95 °C 4 min, 37 cycles (94 °C 50 s, 50 °C 30 s, 72 °C 50 s), 72 °C 7 min.	381
	RpCS.896p RpCS.1233n	GGCTAATGAAGCAGTGATAA GCGACGGTATACCCATAGC		95 °C 4 min, 37 cycles (94 °C 50 s, 51 °C 30 s, 72 °C 50 s), 72 °C 7 min.	337

**Table 2 microorganisms-11-01713-t002:** Details of the number of positive pool for *Rickettsia*, gene amplified and species detected by PCR assay.

Stage/Sex	No. of Pool	No. of Ticks	No +ve	Host +ve	Gene Sequenced				*Rickettsia* spp.
					rRNA	OmpA	OmpB	gltA	
Larva	6	30	0	0	-ve	-ve	-ve	-ve	-
Nymph	14	70	1	Goat	-ve	-ve	*Rickettsia* spp.	-ve	*Rickettsia* spp.
Female	8	40	1	Cow	×	*Rf*	*Rf*	*Rf*	*R. felis*
Male	38	190	2	Goat	*Rf*, *Rr*	*Rr **	*Rr*, *Rf*	*Rr*, *Rf*	*R. felis*, *R. raoultii*

*Rf*: *R. felis*, *Rr*: *R. raoultii*; * Truncated OmpA in *R. felis*, -ve: negative; ×: no PCR amplification.

**Table 3 microorganisms-11-01713-t003:** Details of rickettsial species detected from *H. intermedia* tick, locus amplified, GenBank accession number, percentage of identity with study sequence and cut-off value.

Genes	Taxon	Isolation Source	GenBank Acc. No.	Nucleotide Identity (%)	Closest Match in GenBank	Cut-Off Value (%)
*R. conorii* subsp. *raoultii*						
*16s rDNA*	*R. conorii* subsp. *raoultii*	*H. intermedia*	OP185249	Present study	Present study	rrs > 98.1% (97.7–98.1) with at least one *Rickettsia* species
	*Rickettsia* sp.	Sheep	KT733036	99.32	Spain	
	*R. conorii* subsp. *raoultii*	*D. reticulatus*	MK304546	99.08	Russia	
		*H. asiaticum*	KJ410261	99.08	China	
		Homo sapiens	KY474575	98.86	China	
		*D. silvarum*	CP098324	98.86	China	
*rOmpA*	*R. conorii* subsp. *raoultii*	*H. intermedia*	OK633269	Present study	Present study	Either possess ompA or nucleotide homologies with gltA ≥ 92.7% & *ompB* ≥85.2%
	*Rickettsial* sp.	*D. marginatus*	MG920563	93.38%	Turkey	
	*R. conorii* subsp. *raoultii*	*D. nuttalli*	KU361216	93.38%	Mongolia	
		*I. ovatus*	KX446992	93.38%	China	
		*D. niveus*	JQ792153	93.37%	Tibet	
*rOmpB*	*R. raoultii*	*H. intermedia*	OK633270	Present study	Present study	≥85.8%
	*Rickettsia conorii*	*R. sanguineus*	AF123726	97.39%	France	
	*R. conorii raoultii*	*D.niveus*	JQ792105	96.1%	Tibet	
		*D. everestianus*	JQ792107	96.1%	Tibet	
		Human Blood	KC847318	97.09%	Italy	
*gltA*	*R. conorii* subsp. *raoultii*	*H. intermedia*	OK633271	Present study	Present study	≥92.7%
	*Rickettsial* sp.	Dog blood	MT050445	98.57%	USA	
	*R. conorii* subsp. *raoultii*	*D. marginatus*	KU723493	98.29%	China	
		*D. nuttalli*	MT178338	98%	China	
		Human blood	MH267733	98%	Japan	
*Rickettsia felis*						
*16 s rDNA*	*R. felis*	*H. intermedia*	OP185250	Present study	Present study	rrs > 98.1% (97.7–98.1) with at least one *Rickettsia* species
	*Rickettsia* species	ESTEC HYDRA facility	EU071486	99.54%	Germany	
	*R. felis*	Cultured	CP000053	99.54%	USA	
	*Rickettsia* endosymbiont	*L. bostrychophila*	DQ407743	99.31%	China	
	*R. felis*	*C. felis*	MF303722	99.08	Mexico	
*rOmpA*	*R. felis*	*H. intermedia*	OM675977	Present study	Present study	Either possess ompA or pairwise nucleotide homologies with gltA ≥ 92.7% & *ompB* ≥85.2%
		*C. felis*	AJ563398	100%	Mexico	
		*C. felis*	AY727036	99.6%	USA	
		Infested fleas	EU012496	99.6%	Mexico	
		*L. bostrychophila*	HM636635	99.6%	USA	
		Human Blood	KP318094	99.4%	Bangladesh	
*rOmpB*	*R. felis*	*H. intermedia*	OM675973	Present study	Present study	≥85.8%
		*H. intermedia*	OM675974	Present study	Present study	
		*Ixodes ricinus*	MK301596	100%	Spain	
		Dog Blood	MG451836	99.72%	Italy	
			AF182279	99.72%	USA	
		*R. sanguineus*	JF751024	99.15%	Chile	
*GltA*	*R. felis*	*H. intermedia*	OM675975	Present study	Present study	92.7%
		*H. intermedia*	OM675976	Present study	Present study	
		Booklice	MG818715	99.41%	China	
		Cat flea	MG893575		Malta	
		*C. felis*	MF374381		Austria	
		*Xenopsylla cheopis*	KX446943		Brazil	
		*Ixodes granulatus*	MT847619		Taiwan	

## Data Availability

Not applicable.

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
