# Peer review of "Molecular Evidence of Rickettsia conorii subsp. raoultii and Rickettsia felis in Haemaphysalis intermedia Ticks in Sirumalai, Eastern Ghats, Tamil Nadu, South India"

_microorganisms, 2023, doi:10.3390/microorganisms11071713_

Round 1

Reviewer 1 Report

Dear Editor, The authors of this work have undertaken the study of the prevalence of a particular pathogen in a certain geographical region by using different PCRprotocols in order to detect and identify gene sequences for genetic analysis.Comment 1: The authors could present specific sequence data in their results.Comment 2: Potentially, more samples from different time periods could be selected for analysis. Also, the authors could compare their data using another detection method.Comment 3: the PCR methods were not assayed for sensitivity and specificity.

Minor editing of English language required

Author Response

Dear Editor,

Please see the attachment for reply to reviewer 1

Reviewer 2 Report

Dear authors, below are some comments and suggestions about this article:

- Lines 43 and 45: please remove the parentheses

 - Line 50: replace reference number 8 for this one:

Philippe Parola, Christopher D Paddock, Cristina Socolovschi, Marcelo B Labruna, Oleg Mediannikov, Tahar Kernif, Mohammad Yazid Abdad, John Stenos, Idir Bitam, Pierre-Edouard Fournier, Didier Raoult. Update on tick-borne rickettsioses around the world: a geographic. approach. Clin Microbiol Rev. 2013; 26(4):657-702. doi: 10.1128/CMR.00032-13.

 - Lines 64-66: Please rewrite as follows:

"Rickettsial isolates were recorded from ticks (Amblyomma spp., Haemaphysalis spp. and Rhipicephalus spp.) and fleas (Ceratophyllus fasciatus (rat flea) and C. orientis felis)."

 - Lines 65-66: remove the letter “o” from the word “Rhipicephalous sp.”. The correct is “Rhipicephalus spp.”

 - Add borders to figure 1.

 Line 103: it is important to mention which domestic animals the ticks were collected.

Line 104: change to “during October to December 2020”.

Lines 134, 185, 232 and: write in italics H. intermedia, Rickettsia and Ctenocephalides felis

Lines 195-196: it is necessary to explain the number and stages (adult, nymph or larva) of ticks collected from cows, goats and dogs.

Table 2: add in the legend of the table the meaning of “ve”

Lines 315-317: not write in italics.

Author Response

Dear Editor,

Please see the attachment for the reply to reviewer 2
